# Preparation and Characterization of TiO_2_-PVDF/PMMA Blend Membranes Using an Alternative Non-Toxic Solvent for UF/MF and Photocatalytic Application

**DOI:** 10.3390/molecules24040724

**Published:** 2019-02-17

**Authors:** Ouassila Benhabiles, Francesco Galiano, Tiziana Marino, Hacene Mahmoudi, Hakim Lounici, Alberto Figoli

**Affiliations:** 1Department, Unité de Développement des Equipements Solaires, UDES /Centre de Développement des Energies Renouvelables, CDER, Tipaza 42004, Algeria; benhabiles.ouassila@gmail.com; 2Département génie de l’Environnement, Ecole Nationale Polytechnique, Alger 16200, Algérie; hakim_lounici@yahoo.ca; 3Institute on Membrane Technology (ITM-CNR), Via P. Bucci 17c, 87036 Rende (CS), Italy; t.marino@itm.cnr.it; 4Faculty of Technology, University Hassiba Benbouali of Chlef, Chlef 02000, Algeria; h.mahmoudi@univ-chlef.dz; 5Faculty of Science, University of Bouira, Bouira 02180, Algeria

**Keywords:** polymeric membranes, photocatalytic membranes, blend polymers, TiO_2_ catalyst, organic pollutants, non-toxic solvent

## Abstract

The approach of the present work is based on the use of poly (methylmethacrylate) (PMMA) polymer, which is compatible with PVDF and TiO_2_ nanoparticles in casting solutions, for the preparation of nano-composites membranes using a safer and more compatible solvent. TiO_2_ embedded poly (vinylidene fluoride) (PVDF)/PMMA photocatalytic membranes were prepared by phase inversion method. A non-solvent induced phase separation (NIPS) coupled with vapor induced phase separation (VIPS) was used to fabricate flat-sheet membranes using a dope solution consisting of PMMA, PVDF, TiO_2_, and triethyl phosphate (TEP) as an alternative non-toxic solvent. Membrane morphology was examined by scanning electron microscopy (SEM). Backscatter electron detector (BSD) mapping was used to monitor the inter-dispersion of TiO_2_ in the membrane surface and matrix. The effects of polymer concentration, evaporation time, additives and catalyst amount on the membrane morphology and properties were investigated. Tests on photocatalytic degradation of methylene blue (MB) were also carried out using the membranes entrapped with different concentrations of TiO_2_. The results of this study showed that nearly 99% MB removal can be easily achieved by photocatalysis using TiO_2_ immobilized on the membrane matrix. Moreover, it was observed that the quantity of TiO_2_ plays a significant role in the dye removal.

## 1. Introduction

Achieving a high water quality will be the next major environmental stress issue in the coming decades. Public opinion has evolved recently and demands better water quality and greater respect for the environment. Responding to these requirements the need of adopting new ecologically and environmentally friendly technologies is necessary. Green Chemistry, Green Energy, and Green Environment have been the objective of many researches during the last decades [1,2]. 

In 1987, the World Commission on Environment and Development introduced the concept of sustainable development, and in 1991, the term “green chemistry” was used for the first time by Anastas [3] as part of a special program launched by the Environmental Protection Agency of the United States (USEPA). Green chemistry, however, offers a scientifically-based set of solutions for protecting water quality, human health, and environment. 

The treatment of colored wastewater from textile or dye industry is a serious problem that attracts the attention of many researchers. Dyes’ removal by conventional treatment methods was found to be inadequate because most textile dyes have complex aromatic molecular structures which resist degradation [4]. Although various chemical and physical processes have been employed in the removal of dyes from effluents, they do not lead to a complete degradation of the dyes since most of the employed methods just transfer the contaminants from one phase to another one.

In the last decade, advanced oxidation processes (AOPs) have been growing since they are able to face the issues related to dye decomposition in aqueous systems [5]. The generation of hydroxyl radicals (^•^OH) oxidizing non-selectively various pollutants have improved the degradation of organic compounds compared to conventional methods. The irradiation in the presence of semi-conductors creates a red-ox environment able to destroy organic compounds present in aqueous solutions [6]. Among AOPs, degradation of organic compounds by heterogeneous photocatalysis is considered highly attractive. Titanium dioxide (TiO_2_) is one of the most studied and used photocatalysts thanks to its good photocatalytic activity, high stability, low environmental impact, and low cost [7,8]. This catalyst can work under the form of powder suspended in a slurry or can be immobilized on various supports, such as glass, quartz, stainless steel [9], and recently membranes. 

On the other hand, membrane technology has been already proved to be an alternative technology. The application of membrane processes not only enables high compounds removal efficiencies, but also allows the reuse of water and some of the valuable waste constituents. In the last few years, technical and economical improvements have made the treatment of industrial wastewater by membrane systems even more advantageous. Because of different benefits including high productivity and selectivity, compact and small size equipment, lower cost and energy consumption [10,11,12], the membrane process has found extensive applications in various fields including environment protection [13], petrochemical industry [14], biorefenery [15], desalination, and water treatment [16,17,18,19,20].

Despite the effectiveness of the membrane process, one of the main hurdles is represented by the decline of permeate flux due to the accumulation of various materials on the membrane surface. This phenomenon, caused by fouling and concentration polarization processes, leads to a reduction of membrane performance. Several approaches have been proposed so far in order to limit and mitigate fouling phenomenon during the treatment of wastewater [21,22,23,24,25]. Among these solutions, a promising method is coupling photocatalysis and membrane techniques [26]. A synergistic combination of photocatalysis and membrane filtration may provide a solution to enhance organics degradation and membrane permeability. 

The combination of the membrane separation with the photocatalysis process is called photocatalytic membranes reactor (PMRs) and it represents an efficient, low cost and environmentally friendly technology with a great potential in water/wastewater treatment. In PMRs configurations, the catalysts can be immobilized on the membrane surface, entrapped in the membrane matrix or suspended in water [27,28]. Most of the works have focused on the use of TiO_2_ photocatalysts suspended in water [29]. However, the main technical barriers to its commercialization are represented by the fact that the photocatalysts need to be separated from the suspension causing an increase in the operating costs and a decrease of the treatment efficiency [30]. A promising solution to this problem is to use TiO_2_ photocatalysts entrapped within the membrane matrix.

A great number of organic polymers are commercially available for the preparation of ultrafiltration (UF) membranes by phase inversion method. Among these polymers, poly (vinylidene fluoride) (PVDF) is a semi-crystalline polymer with widespread applications thanks to its excellent properties such as good mechanical strength, stability against harsh chemicals and good thermal stability [31,32]. However, the high hydrophobic nature and the low fouling resistance of PVDF membranes (due to protein adsorption on their surface) limit their use in water treatment processes. Many studies have been devoted in improving or varying PVDF membranes properties adopting several techniques such as physical blending, chemical grafting, and surface modifications [33,34,35]. 

Among these various methods, the blending with various polymers has the advantage of easy preparation by the phase inversion technique. The addition of a second polymer to the casting dope solution is known to be very effective on the modification of membranes structure and characteristics, such as the formation of macrovoids, which produces different behaviors in filtration [33].

PVDF is highly miscible with oxygen containing polymers, thanks to the interaction between the fluorine atoms and carbonyl groups of the partner polymer [36,37]. Several pairs of blends have been investigated, such as PVDF/poly(vinylpyrrolidone) (PVP), PVDF/ poly (ethylene glycol) (PEG), PVDF/ poly(vinylacetate) and PVDF/poly(methylmethacrylate) (PMMA) [38]. Among these polymers, PMMA is one of the most interesting due to its good compatibility with PVDF by simultaneous precipitation in a water bath all over the concentration range [37,39,40,41,42,43]. The produced blend membranes exhibited improved hydrophilicity and water permeation fluxes with respect to pure PMMA membranes [44]. The blend of PVDF with PMMA was investigated by several authors due to the possibility to combine the benefits of both polymers (improving mechanical resistance, processability, and ionic conductivity) [45,46,47,48]. However, a systematic study aiming to evaluate the effect of preparation parameters (such as evaporation time, polymer concentration and presence of pore formers) is still missing. Moreover, the potential photocatalytic activity of PVDF/PMMA membranes loaded with TiO_2_ nanoparticles is an unexplored area.

Considering that photocatalytic composite membranes using alternative solvents have not been thoroughly investigated, the preparation of nano-composite blend PVDF/PMMA membranes containing TiO_2_ using a more compatible solvent is reported and discussed in this work for the first time. 

In order to preserve the environment from the risks posed by conventional toxic solvents [49], triethyl phosphate (TEP) has been, therefore, used as much less harmful alternative solvent, in accordance to the concept of sustainable development and green chemistry principles [50]. TEP is not teratogenic, carcinogenic, or mutagenic, and as stated in its Safety Datasheet “contains no components considered to be either persistent, bioaccumulative and toxic, or very persistent and very bioaccumulative at levels of 0.1% or higher [51]”.

So far, there are only five articles concerning the use of TEP, as only solvent, for the production of PVDF membranes applied in membrane distillation but none of them dealing with water purification [52,53,54,55,56]. TiO_2_–PVDF/PMMA/TEP membranes were prepared via the phase-inversion method. Scanning electron microscopy (SEM) was used to characterize the surface and cross-section morphology of the membranes. Pore size, pure water permeability (PWP), mechanical strength, porosity, and contact angle measurements were also carried out. The photocatalytic activity of the membranes was evaluated by following the methylene blue (MB) degradation in water under UV light irradiation. 

## 2. Results and Discussion

### 2.1. Cloud Point Measurement

This work describes, for the first time, the preparation of PVDF/PMMA membranes using TEP as a solvent. However, the ability of TEP to dissolve PVDF has been already studied in previous works through the comparison of their Hansen solubility parameters [52,56]. The solubility parameter of TEP (22.2 MPa^1/2^), in fact, is very close to the one of PVDF (23.2 MPa^1/2^) confirming that a PVDF-TEP solution is thermodynamically favored. 

A large number of studies have focused on the phase diagrams of PVDF solutions in relation to the cloud point temperature at which a clear polymer solution becomes more viscous and cloudy [57,58,59]. Opacity can occur in a polymer solution due to the separation between lean and polymer-rich phases, and this process is known as liquid–liquid phase separation (L–L). In our PVDF/PMMA/PET systems, the solutions turned cloudy and then started to form gels at the sol-gel transition temperature, as described in Figure 1.

The sol-gel transition temperatures measured for TEP as a function of the concentration of PVDF/PMMA are presented in Figure 2. The obtained results show that the sol-gel transition temperature increased at higher PVDF/PMMA concentrations and it was ranging from 43 to 55 °C. This can be explained by the fact that a higher temperature is required to keep the polymer/solvent system homogeneous when the concentration of polymer is increased.

Same trend was observed by Sawada et al. [59] for the preparation of PVDF membranes via TIPS using three non-toxic citrate-based solvents. For all the solvents evaluated in this study, the sol-gel transition temperature increased with the increase of PVDF concentration.

### 2.2. Characterization of the Membranes

The effect of some crucial parameters like evaporation time, amount of polymer, additives and quantity of TiO_2_ was investigated. The influence they played in membrane morphology, performance and properties was investigated and discussed.

#### 2.2.1. Effect of Evaporation Time

The time required to evaporate the solvent from the polymer solution during the phase-inversion process is known as evaporation time. This is one of the main parameters in membrane manufacturing which can be tuned in order to produce the desired morphology in porous membranes. The membrane morphology has a significant effect on the structural properties and performance of UF/microfiltration (MF) membranes. The influence of evaporation time was, therefore, investigated for the membranes M1, M2, and M3 whose composition is reported in Table 1.

Flat sheet membranes were prepared using the same dope solution and three different evaporation times, ranging from 0 to 5 min, were considered. The membranes, after casting, were immersed immediately in the coagulation bath (membrane M1, NIPS) or exposed for 2 min 30 s or 5 min to a relative humidity before the precipitation of the polymer in water (membrane M2 and M3, NIPS-VIPS). 

Figure 3 show SEM images of top, bottom and cross-section of the different investigated membranes. Due to the good miscibility of PVDF and PMMA, all the prepared membranes did not show any phase separation. The morphology of M1 membrane revealed an asymmetric structure with a homogeneous and uniform dense layer overlying a porous and thicker structure. The denser top layer is the result of the fast solvent/non-solvent demixing rate occurring for the M1 membrane prepared by NIPS as a consequence of the immediate immersion of the cast film in the water coagulation bath. The exposure of the nascent films (M2 and M3 membranes) to humidity, on the contrary, slowed down the precipitation process fostering the formation of a more porous surface visible in Figure 3d,g.

All membranes presented a porous and sponge-like structure along the cross section (for higher magnification see the Appendix A). M3 membrane (Figure 3i), however, showed an asymmetric structure: a thin layer at the top surface in the form of macrovoids and a sponge-like structure across the rest of the membrane.

The evaporation time (different from 0) significantly changed the morphology of the membrane surface and also the nature of the sponge-like structure. Similar results were observed by A. Ali et al. [60] which reported that the effect of evaporation time considerably altered the morphology and structure of membranes made from a mixture of polysulfone/cellulose acetate phthalate/polyvinylpyrrolidone (PSf/CAP/PVP). The change in the membrane morphological structure with increasing evaporation time is due to the movement of the volatile solvent on the surface of the spread film. Increasing the evaporation time, more solvent was removed from the surface of the membrane. In this phenomenon, a homogeneous and more concentrated nascent skin layer was formed due to the coalescence and melting of the separated dry phase structure. This type of skin layer has a high resistance to solvent and non-solvent mass transfer between the coagulation bath and the inner region of the membrane during the wet phase inversion process [61].

The trend of PWP of the membranes (presented in Figure 4) confirmed what observed in SEM pictures and it was also in agreement with the characterization tests reported in Table 1. 

The PWP, in fact, increased from 29 L/m2 h bar to 226 L/m2 h bar by increasing the evaporation time of the membrane. This was due to an enhancement in membrane pore size (from 0.08 μm for M1 to 0.32 μm for M3) and porosity (from 80 % for M0 to 87 % for M3). The increase in pore size during the VIPS process is a very well known phenomenon. The higher exposure time, in fact, lowers solvent-non-solvent demixing delaying polymer precipitation and, thus, favoring an increase in membrane pore size and porosity [62]. The membrane pore size was in the range of MF/UF.

The mechanical properties of the membranes, however, decreased by increasing the evaporation time. The Young’s modulus dropped from 87 to 55 N/mm^2^ as a result of the increased porosity of the membranes. This is in agreement with what generally observed in literature. The presence of macrovoids (as in M3) and the increase of the overall void fraction (as in M2 and M3) represent weak points into the membrane leading to a decrease in membrane mechanical resistance [63]. 

All the three membranes presented a contact angle higher than 90° (from 98 to 123°) as a consequence of the polymers used for their fabrication. PVDF and PMMA, in fact, are both hydrophobic polymers conferring to the membranes a hydrophobic moiety. The difference in contact angle from M1 to M3 membrane can be justified by their different surface morphology. M2 and M3 membranes, in fact, presented a porous surface in comparison to M1 characterized by a denser and more compact topography. Besides surface topography, membranes having higher pore size shows also higher surface roughness [64,65,66,67]. 

The higher roughness of porous membranes can be responsible of the hydrophobicity reinforcement [68] according to the phenomenological model proposed by Wenzel [69].

The intermediate evaporation time of 2.5 min (M2 membrane) was considered and kept constant for the preparation of the following M4–M8 membranes. 

#### 2.2.2. Effect of Polymer Content

The polymer concentration is one of the fundamental parameters in the manufacturing of membranes strongly affecting their morphology and their retentive capacity. In this study, three different concentrations of the PVDF-PMMA blend were considered: 10, 12, and 14 wt%. All the membranes were prepared by keeping constant the evaporation time at 2.5 min (M2, M4, and M5 membranes whose complete composition is reported in Table 6).

Figure 5 illustrates top surface, bottom surface and cross-sections of the investigated membranes. M2 and M4 membranes, prepared with 12 and 10 wt% of polymers concentration, respectively, presented a very porous surface for both top and bottom side (as showed in Figure 5a,b,d,e). The membrane surface of M5 prepared with 14 wt% of polymers concentration (Figure 5g), however, was quite different. Increasing the polymer concentration, in fact, a less-porous surface was observed. The reduction of cavities as a consequence of polymer concentration increase is a very well documented phenomenon [63,70]. High polymer concentrations, in fact, slow down the precipitation process of the membrane leading to the formation of a membrane with a denser top layer and lower pore size [71]. The cross-section of the membranes revealed a sponge-like structure (for higher magnification see the Appendix A). 

As indicated in Table 2, the highest polymer concentration (14 wt%) caused, as expected, an increase in the viscosity of the dope solution (from 388 to 2133 cP). This increase slowed down the growth of the porous structures, resulting in a reduction of the pore size and consequently in a reduction of the porosity (from 85 to 82%). 

The cross-section of all the three membranes (Figure 5c,f,i) was mainly represented by a sponge-like structure even if in M5 membrane is visible the denser skin layer.

All the membranes exhibited a hydrophobic nature as a consequence of the polymers used for their preparation. Even in this case, the lower contact angle exhibited by M5 membrane can be related to its smoother and more compact surface in comparison to M2 and M4. 

#### 2.2.3. Effect of Additives

Several studies have been carried out showing the effect of additives on the structure and the performance of UF PVDF membranes [72]. The performance and the membrane structure can be controlled by the composition of the dope solution and other membrane preparation conditions. In Table 3, the effect of additives (PEG-200 and PVP-K17) was studied for two membranes (M0 and M2) prepared at the same polymers concentration (12 wt%) and same evaporation time (2.5 min). M0 membrane did not contain any additive, while M2 membranes contained 25 wt% of PEG200 and 5 wt% of PVP-K17 both very well known to act as porogens.

The additives used did not greatly affect the overall wettability of the membrane. PVP and PEG are hydrophilic agents which can improve the hydrophilic nature of the membrane. However, during the membrane post-treatment they are washed away and removed from the membrane matrix being both soluble in water. This process can, therefore, nullify their hydrophilic behavior on the membrane. The different values in contact angle between M0 and M2 can therefore be attributed to the different morphology exhibited by both membranes as can be seen in Figure 6.

The presence of additives, as expected, promoted an increase in porosity (from 78 to 82%) and in membrane pore size (from 0.07 to 0.14 μm).

Additives with a concentration ranging from 5 to 20 wt%, are also generally responsible of membrane thickness increase as a consequence of dope solution viscosity increase and the formation of a more compact structure [73,74,75]. The addition of the additives, in fact, induced an increase in the dope solution viscosity (from 228 to 645 cP) (almost 3 times higher) and the formation of a denser kin layer (Figure 6). As can be seen in cross-section pictures (Figure 6c,f), the morphology of the membrane varied from a spherulitic structure (in case of M0) to a sponge-like structure (in case of M2). SEM magnification of membranes top layer is shown in Appendix A.

The Young’s module decreased from 90 to 81 N/mm^2^ as a consequence of additives addition in the membrane matrix due to the increase of the overall void fraction.

#### 2.2.4. Effect of TiO_2_ Amount

The TiO_2_ concentration generally ranges between 0 and 5 wt% by according to various studies reported in literature [45,76,77,78,79,80,81,82,83]. In the present research, membranes containing different amounts of TiO2 (0.12, 0.25 and 0.5 wt%) were prepared. Table 6 reports the different formulations of the dope solutions used for the membranes prepared at different TiO2 concentrations (M6, M7, and M8) and at the same evaporation time of 2.5 min.

The dispersion of inorganic nanoparticles in the polymer matrix is an important characteristic of organic/inorganic composites. The SEM-BSD analysis was used to map the inter-dispersion of TiO2 particles on the surface and membrane matrix. Figure 7 shows SEM-BSD images of the surfaces and cross-sections of TiO2-PVDF/PMMA nano-composites membranes prepared by phase inversion compared with a control sample without catalyst (M2). 

No clear particle aggregation was observed in the cross sections of the composites, indicating that the TiO2 nanoparticles were well dispersed in the polymer matrix. These results are similar to the ones reported by Zho et al. [82] for PVDF/PMMA/TiO2 composite membranes prepared by in situ polymerization. Even if some areas of the membranes contained TiO2 aggregations, as shown in Figure 7(c4), TiO2 particles were generally well dispersed not only on the surfaces but also throughout the membrane cross-section (Figure 7(c2–4)). 

TiO2-PVDF/PMMA nano-composites membranes were characterized by pore size diameter, contact angle, porosity, and mechanical strength (Table 4). By increasing the amount of TiO_2_, the hydrophilicity of the membrane was enhanced as a consequence of the hydrophilic moiety of the nanoparticles [84]. Moreover, the addition of TiO2 led to a general improvement in membrane Young’s modulus value in comparison to the unloaded membrane M2. As reported in several works [83,85,86], in fact, the introduction of TiO2 nanoparticles enhances the mechanical resistance of the membranes by acting as cross-linkers between the polymer chains increasing their rigidity and, as a consequence, their mechanical resistance. The TiO2 nanoparticles can, in fact, interact via hydrogen bond between the hydroxyl group present on the surface of TiO2 and oxygen groups of PMMA [87,88].

The pore size of the membranes increased as the increase of TiO2 (from 0.14 μm for M2 to 0.42 μm for M8). The increase in pore size after the addition of TiO_2_ can be related to the formation mechanism of the membranes. The TiO_2_ nanoparticles have a hydrophilic nature and this could promote the attraction and penetration of water vapor from the environment during the VIPS process favoring the increase in membrane pore size [86].

### 2.3. Photocatalytic Performance

Before carrying out the photocatalytic tests with MB, the resistance of the blend membranes to UV irradiation was evaluated in order to exclude any possible photodegradation of the polymer matrix. The tests were carried out with the M8 membrane, containing the highest concentration of TiO_2_, by measuring its starting PWP. Then, the membrane sample was exposed for 3 h to UV light irradiation. After this period, the PWP was measured again. No changes in the recorded values of PWP (about 780 L/m^2^ h bar), before and after the exposure to UV light irradiation, were observed (see Appendix A). From the constancy of membrane performance, the resistance of the membrane to UV light was, therefore, assessed. In case of photodegradation of the polymer, in fact, an increase in water permeability should have been observed. The higher PWP of M8 membrane in comparison to M1, M2, and M3 can be attributed to its higher pore size and lower contact angle as a consequence of TiO2 addition.

The photocatalytic activity of TiO2-PVDF/PMMA prepared membranes was studied by evaluating the photodegradation of the model dye MB. The photocatalytic performance of the membranes was evaluated by measuring the concentration of MB during 300 min of UV lamp (Zp type 500 W) irradiation.

In order to examine the effect of TiO2 catalyst incorporated in photocatalytic membranes matrix, a membrane without TiO2 was also used (M2) under the same operating conditions. The variation in MB concentration as a function of UV irradiation time in the presence of different membranes is shown in Figure 8.

According to the obtained results, the membranes loaded with TiO_2_ exhibited significant photocatalytic activity in comparison to the pristine one (M2 membrane). By increasing the TiO_2_ content, the photodegradation of MB increased. M6, M7, and M8, in fact, showed the 86%, 95%, and 99% MB decomposition rates after 330 min UV lamp irradiation, respectively. 

The membrane without catalyst M2 gave 51% of MB removal due to photolysis and membrane adsorption phenomenon. The maximum MB removal rate of 99% was achieved after only 150 min of treatment using the M8 membrane corresponding to the maximum amount of TiO_2_ used for this study.

The curves of the reduced concentration have an exponential appearance; the kinetics of degradation of the BM is of pseudo-first order. It is well established in the literature that for a low initial concentration C_0_, the degradation rate of organic pollutants follows the Langmuir–Hinshelwood (L–H) law. This model allows us to determine the reaction rate constant from the curve plot Ln (C/C_0_). The plot is a line that passes through the origin; the apparent constant kapp is the slope of this line. The values of the kinetic constants, the initial rates r0, the degradation rate X% and the time of half- reaction t1/2 are calculated and grouped in Table 5.

The half-reaction time is a very important parameter for the explanation of the elimination kinetics if we consider that t_1/2_, is the time necessary to degrade half of the MB quantity. Therefore, the smallest half-reaction time corresponds to the best rate of degradation.

The half-reaction time increases as the reaction rate decreases. The evolution of t_1/2_ as a function of the TiO_2_ content shown in Figure 9 makes possible to deduce that the difference in photoactivity is clearly reduced for the lowest TiO_2_ contents.

These preliminary results can be considered as a starting point for a future far-reaching investigation of membranes photocatalytic activity directly measured during the operation time of the membrane (during filtration) considering, also, different organic pollutants. 

## 3. Experimental

### 3.1. Chemicals

PVDF (Solef^®^ 6010, ~322,000 g/mol) supplied by Solvay Specialty Polymers (Bollate, Italy) and PMMA purchased by Sigma-Aldrich ~350,000 g/mol were used as polymers, TEP (Sigma-Aldrich, Milan, Italy) was used as an alternative solvent. PVP (K17, BASF, Ludwigshafen, Germany; M.W. ~10,000 g/mol) and PEG (PEG-200, Sigma-Aldrich, Milan, Italy; M.W. ~200 g/mol) were added, as pore-forming agents, to the polymeric solution. TiO_2_ (Aeroxide-Degussa P25) photocatalyst was dispersed in dope solution. As reported by the supplier, the TiO_2_ nanoparticles have a primary mean diameter of about 21 nm, with a density of about 4 g/cm^3^ with a predominance of the anatase form. Bi-distilled water was used as a non-solvent for polymer precipitation.

### 3.2. Membrane Preparation

PVDF/PMMA and TiO_2_-PVDF/PMMA composite membranes were prepared via the phase-inversion method. The dope solution was prepared by mixing appropriate amounts of PVDF, PMMA and additives in TEP at 100 °C and continuously stirred for more than 12 h until a homogeneous solution was obtained. Different amount of TiO_2_ particles were added to the mixture. The solution was then cast on a glass plate using a manual casting knife (Elcometer 3700 Film Applicator Blade with Tank, Elcometer Instrument GmbH, Aalen, Germany) with a set thickness of 350 μm.

The nascent membranes were either immediately immersed in the coagulation bath (NIPS) or exposed for a variable time (5 and 10 min) under controlled relative humidity (RH, 65%) and temperature (25 °C) in a climatic chamber (DeltaE srl, Rende (CS), Italy) before their complete coagulation into an aqueous coagulation bath (NIPS coupled with VIPS). 

In order to remove solvent and additives traces, the formed membranes were then washed with hot water (60 °C) three times consecutively. Then membranes were stored in a water bath until being used. In Table 6, a list of the produced membranes with details of dope composition and evaporation time used is reported.

### 3.3. Membrane Characterization

#### 3.3.1. Viscosity Measurements

The viscosity measurements of the cast solutions were conducted using a DVIII-Brookfield viscometer at 90 °C.

#### 3.3.2. Contact Angle Measurements

Water contact angles θ for the top and bottom sides of the membranes were measured by an optical tensiometer (CAM100 Instrument, Nordtest srl, GI, Serravalle Scrivia (AL), Italy) via the drop method. A 3-μL bi-distilled water droplet was placed on the membrane surface. Each value was obtained immediately after dropping water. To minimize the measurement error, a total of five replicates were taken and the average was calculated. 

#### 3.3.3. Porosity Measurement

The membrane porosity is defined as the pore volume divided by the total volume of the membranes. Each membrane sample was weighed and subsequently submerged in a container filled with kerosene and stored for 24 h [89]. The test was performed three times and the porosity value was calculated. The porosity (ε) was estimated using Equation (1):(1)$$\varepsilon \left(\%\right)=\frac{V_{Pore}}{V_{Total}}=\frac{(w_{W}-w_{d})/\rho_{s}}{\left(w_{w}-w_{d}\right)/\rho_{s}+\left(w_{d}/\rho_{p}\right)}\times 100$$
where:
$w_{w}$ is the weight of the wet membrane,$w_{d}$ is the weight of the dry membrane,$\rho_{s}$ is the kerosene density,$\rho_{p}$ is the polymer density.

#### 3.3.4. Pure Water Permeability (PWP)

The PVDF/PMMA and PVDF/PMMA/TiO_2_ membrane permeation performance experiments were conducted measuring the pure water permeability (PWP). PWP was measured at 25 °C using a laboratory cross flow set-up. Water was fed through the membrane (area of 0.0008 m^2^) by means of a gear pump (Tuthill Pump Co., Concord, CA, USA). PWP was, then, calculated using Equation (2): (2)$$PWP=\frac{Q}{AtP}$$
where:
*Q* is the volume of the permeate in liters,*A* is the membrane surface area (m^2^),*t* is the permeation time (h),*P* is the applied pressure used (bar).

A minimum of three membrane samples were tested and the tabulated results show the average values.

#### 3.3.5. Pore Size

The pore size of prepared membranes was evaluated by the liquid-gas displacement technique using a Capillary Flow Porometer (CFP- 1500 AEXL, Porous Materials Inc., Ithaca, NY, USA). Membranes were soaked in Porewick^®^ used as a wetting liquid (superficial tension 16 dyne/cm) for 24 h to be completely wetted. Nitrogen was gradually allowed to flow into the membrane by increasing its pressure during time. Gas pressure as well as permeation flow rates across the dry membrane were registered, allowing the final mean pore size calculation. 

#### 3.3.6. Microscopy

The membrane morphology was evaluated by using a Zeiss-EVO Ma10 instrument (Zeiss, Oberkochen, Germany) scanning electron microscope (SEM). Cross sections of the membranes were prepared by fracturing the membranes in liquid nitrogen. All of the membranes were coated with a thin layer of gold before scanning in order to make the samples conductive. SEM-BSD mapping was used to monitor the inter-dispersion of TiO2 in the membrane matrix.

#### 3.3.7. Mechanical Strength

Young’s modulus was measured using Zwic/Roell test unit (Ulm, Germany). For each membrane, five samples (5 cm × 1 cm) were tested, the tabulated results report the average values.

### 3.4. Experimental Set-up for Photocatalytic Performance

A batch photocatalytic reactor consisted of a pyrex glass tank with a piece of membrane was placed inside a shielded chamber with UV lamp mounted on the top (lamp emission from 180 nm to visible light–500 W–Purchased from Helios Italquarz s.r.l., Cambiago (MI), Italy). 

MB degradation experiments were carried out for 10 μmol feed solution. Flat sheet membranes containing different quantities of TiO_2_ (0.12, 0.25, and 0.5 wt%) were prepared by phase inversion technique. Membranes with the same surface dimension (4 cm × 4 cm) were immersed in 400 mL feed solution as described in Figure 10.

Adsorption tests were carried out in the dark for 30 min before lighting the UV lamp for all experiments. The decomposition of the MB was carried out at ambient temperature in the batch reactor described in Figure 10. The photocatalytic tests were carried out by immersing M2, M6, M7, and M8 membranes in 500 mL of BM solution with an initial concentration 10 μmol/L under UV lamp irradiation.

The photocatalytic degradation of MB for both initial concentration and irradiated samples was determined by UV–Vis spectrophotometer analysis (Shimadzu UV-1601, Kyoto, Japan) following the decrease of dye absorption as a consequence of its degradation. A calibration curve of MB solution obtained at 664.5 nm wavelengths for different concentrations was prepared in order to correlate the concentration of MB at different reaction times by converting the absorbance of the sample to MB concentration.

## 4. Conclusions

In this study porous PVDF/PMMA membranes and TiO_2_-PVDF/PMMA blend membranes were prepared by the phase inversion method using TEP as an alternative solvent by evaluating the effect of different preparation and operation conditions. The effects of evaporation time, additives and polymer content on morphology, mechanical stability and membrane performance were, in fact, systematically investigated. Furthermore, the effect of the TiO_2_ addition to the PVDF/PMMA membranes was evaluated in terms of membrane properties and photocatalytic activity.

The morphology and the physical properties of the prepared membranes were found to be dependent both on the composition of the casting dope solution and on the phase inversion technique adopted (NIPS or VIPS/NIPS). The results can be, therefore, summarized as follows: The membranes produced via the NIPS-VIPS procedure showed a significant change in the morphology of the membrane, from a dense to a porous structure in the range of UF/MF. The pore size was also improved with a subsequent decrease of membranes mechanical properties.The presence of hydrophilic additives such as PVP and PEG induced a morphological evolution from a very dense structure to a porous one. The additives enhanced the porosity of the membrane favoring the formation of a more open structure with an increase in pore size.The addition of different amount of TiO_2_ led to an increase in membrane pore size, improving hydrophilic nature and mechanical properties of the membranes.The photocatalytic membranes developed showed promising results in terms of MB photodegradation (up to 99%) making these membranes potential candidates for the treatment and purification of wastewater.

## Figures and Tables

**Figure 1 molecules-24-00724-f001:**
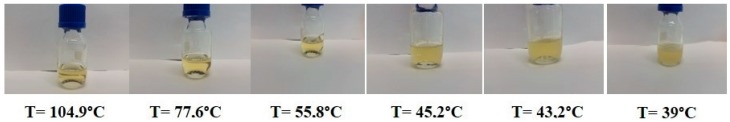
Visible phase change for a 12% polymer concentration solution.

**Figure 2 molecules-24-00724-f002:**
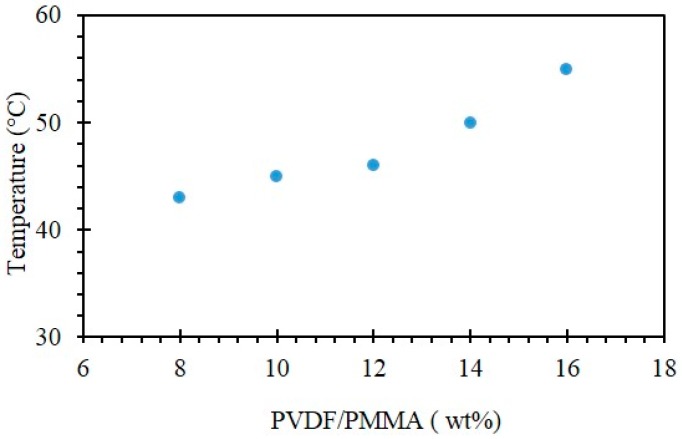
Sol-gel transition temperature for different concentrations of PVDF/PMMA (poly (vinylidene fluoride/poly (methylmethacrylate)).

**Figure 3 molecules-24-00724-f003:**
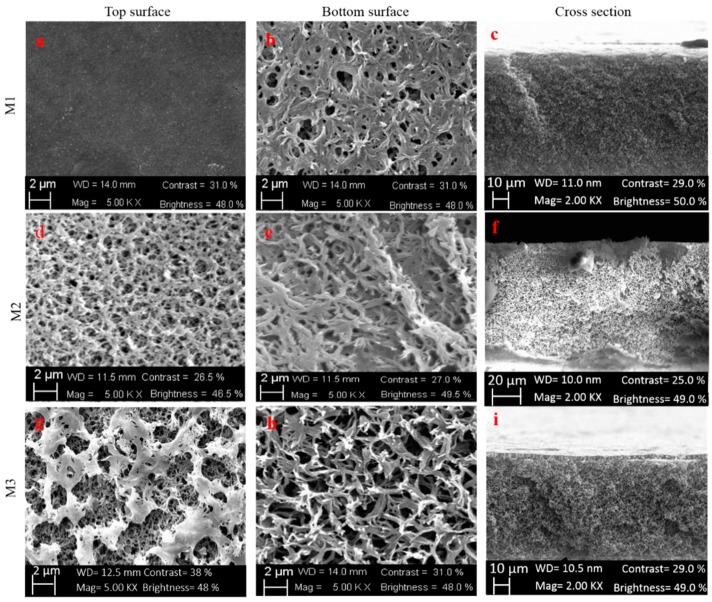
SEM images: top surface, bottom surface and cross-sections of membranes M1, M2, and M3.

**Figure 4 molecules-24-00724-f004:**
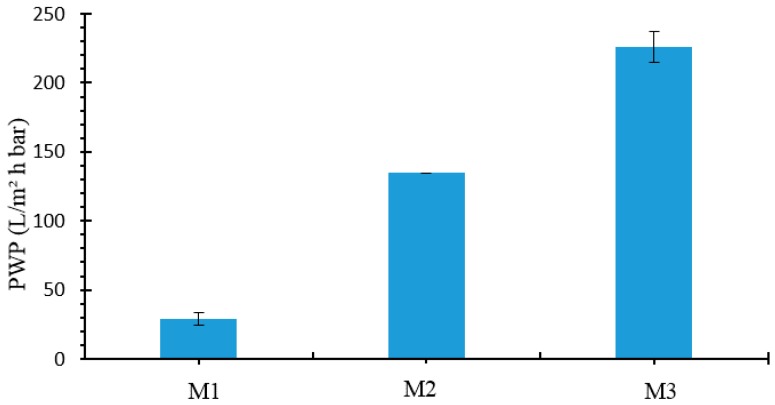
Pure water permeability (PWP) of membranes with different evaporation time.

**Figure 5 molecules-24-00724-f005:**
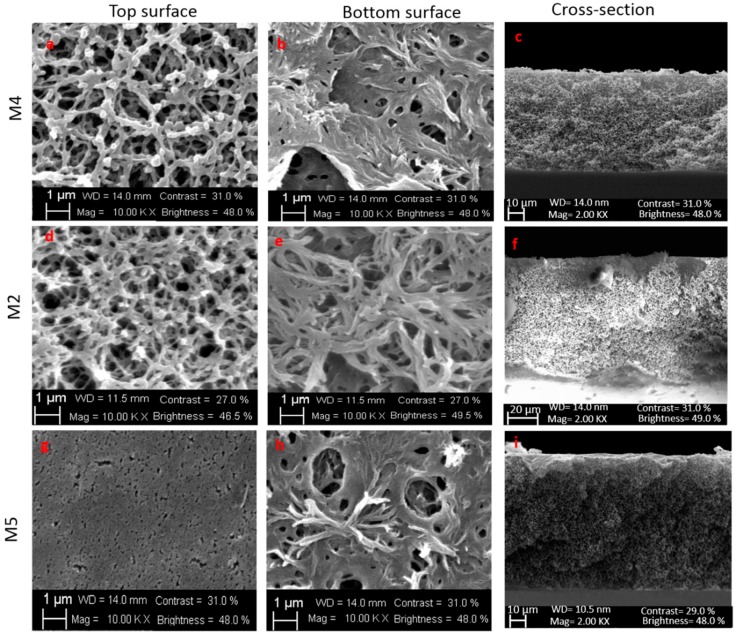
SEM images: top surface and bottom surface of membranes M4, M2, and M5.

**Figure 6 molecules-24-00724-f006:**
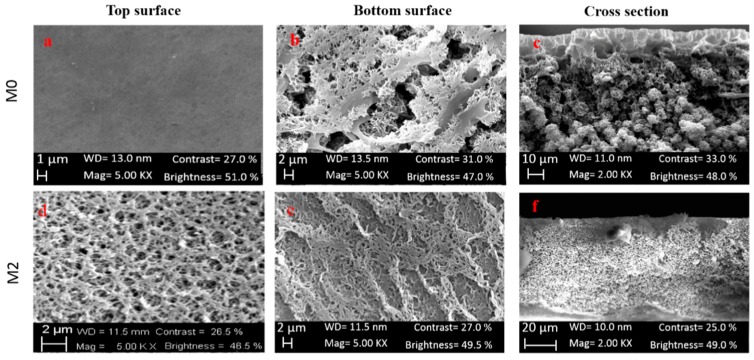
SEM images: top surface, bottom surface and cross-section of membranes M0 and M2.

**Figure 7 molecules-24-00724-f007:**
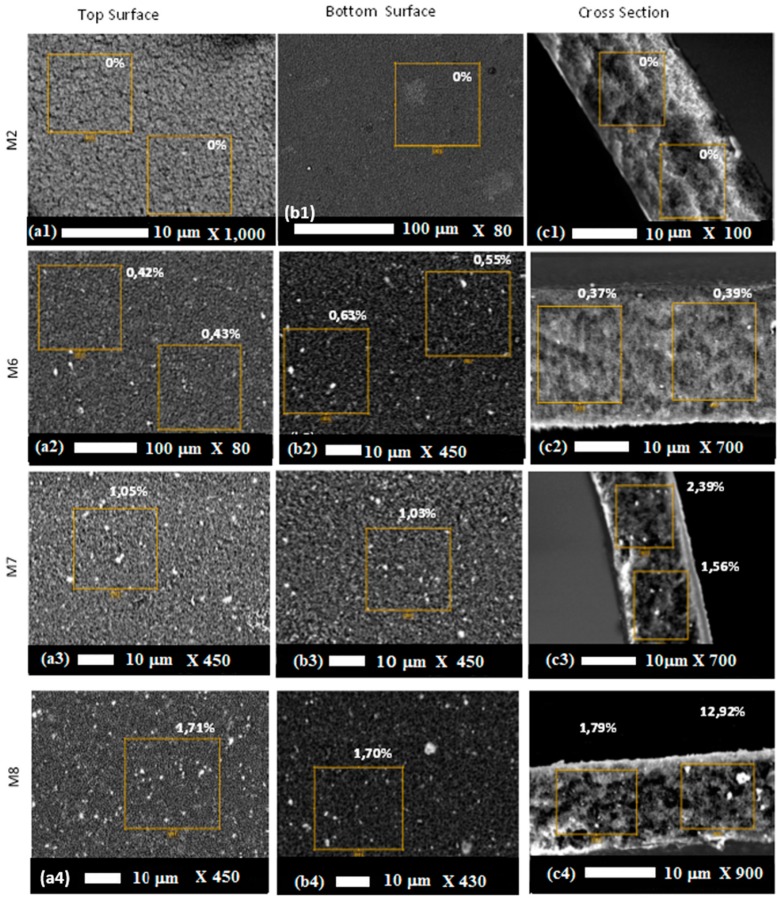
SEM-BSD (backscatter electron detector) pictures for TiO_2_ particles distribution on top surfaces (**a**), bottom surfaces (**b**) and cross sections (**c**) of M2, M6, M7, and M8 membranes.

**Figure 8 molecules-24-00724-f008:**
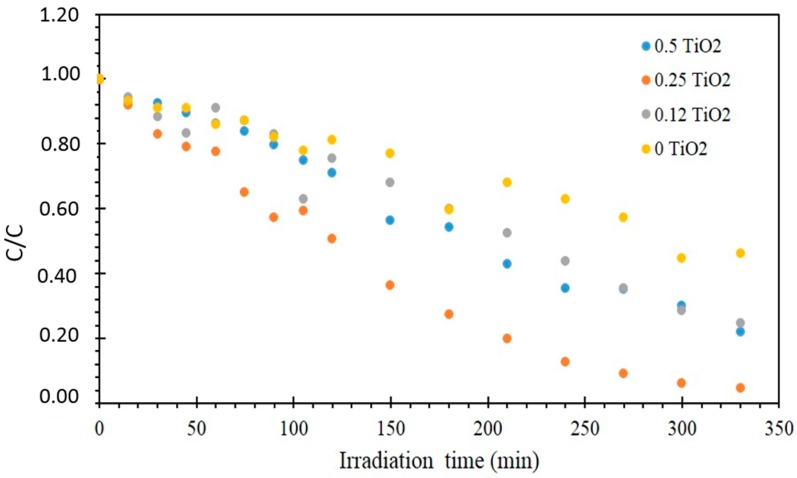
MB degradation using membranes with different amount of TiO_2_ under UV irradiation.

**Figure 9 molecules-24-00724-f009:**
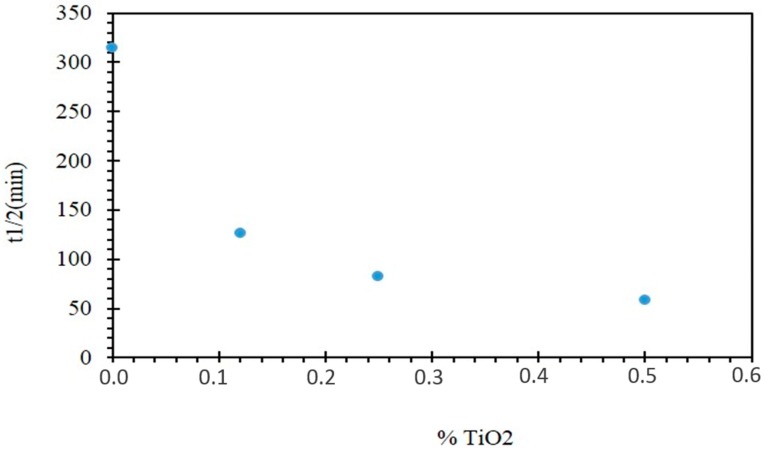
Evolution of t_1/2_ as a function of the TiO_2_ content incorporated into the membrane matrix.

**Figure 10 molecules-24-00724-f010:**
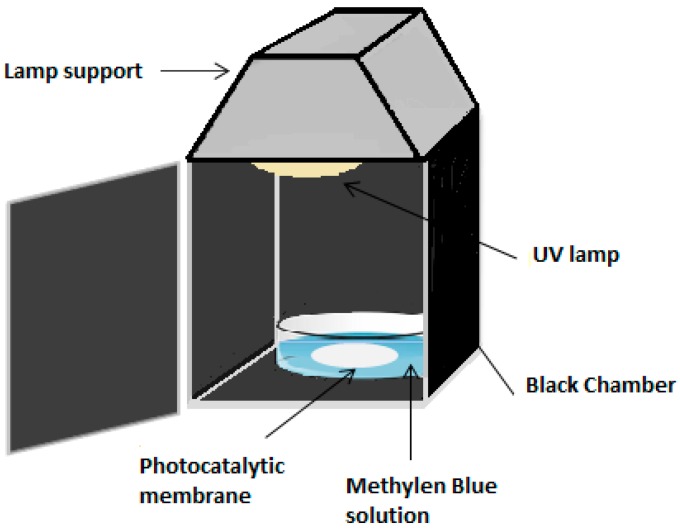
Schematic picture of the photocatalytic set-up.

**Table 1 molecules-24-00724-t001:** Characterization of the M1–M3 membranes.

Membrane Code	Porosity (%)	Mean Flow Pore Diameter (µm)	Young’s Modulus N/mm²	Contact Angle (°)
**M1**	80 ± 1	0.08 ± 0.05	87 ± 1	98 ± 1
**M2**	82 ± 1	0.19 ± 0.1	81 ± 1	120 ± 3
**M3**	87 ± 1	0.32 ± 0.2	55 ± 2	123 ± 3

**Table 2 molecules-24-00724-t002:** Membrane characterization of membranes prepared at different polymer content.

Membrane Code	Viscosity (cP)	Thickness (mm)	Porosity (%)	Mean Flow Pore Diameter (µm)	Young’s Modulus (N/mm²)	Contact Angle (°)
**M4**	388	0.056 ± 0.012	85 ± 1	0.40 ± 0.2	84 ± 2	124 ± 1
**M2**	645	0.072 ± 0.008	82 ± 1	0.19 ± 0.1	81 ± 1	120 ± 3
**M5**	2133	0.077 ± 0.006	82 ± 1	0.18 ± 0.2	142 ± 12	111 ± 1

**Table 3 molecules-24-00724-t003:** Characterization of membranes prepared without and with additives.

Membrane Code	Viscosity (cP)	Thickness (mm)	Porosity (%)	Mean Flow Pore Diameter (µm)	Young’s Modulus (N/mm)	Contact Angle (°)
**M0**	228	0.061 ± 0.004	78 ± 1	0.07 ± 0.05	90 ± 1	110 ± 2
**M2**	645	0.072 ± 0.008	82 ± 1	0.14 ± 0.1	81 ± 1	120 ± 3

**Table 4 molecules-24-00724-t004:** Membrane characterization for different amounts of TiO_2_.

Membrane Code	Thickness (mm)	Porosity (%)	Mean Flow Pore Diameter (µm)	Young’s Modulus (N/mm²)	Contact Angle (°)
**M2-0Ti**	0.072 ± 0.008	82 ± 1	0.14 ± 0.1	81 ± 1	120 ± 3
**M6-0.12Ti**	0.063 ± 0.003	80 ± 1	0.35 ± 0.2	114 ± 5	102 ± 4
**M7-0.25Ti**	0.068 ± 0.011	81 ± 1	0.41 ± 0.25	99 ± 7	98± 4
**M8-0.50Ti**	0.055 ± 0.008	83 ± 1	0.42 ± 0.15	104 ± 3	93 ± 5

**Table 5 molecules-24-00724-t005:** Values of apparent constants (k_app_), time of half- reaction (t_1/2_) and initial velocities (r_0_) for different amounts of TiO_2_.

Membrane Code	TiO_2_ (%)	k_app_ (min^−1^)	r_0_ (mg/L·min)	t_1/2_ (min)	X (%)	R²
**M2**	0	0.0022	0.022	315	51	0.928
**M6**	0.12	0.0055	0.051	126	86	0.979
**M7**	0.25	0.0084	0.074	83	95	0.952
**M8**	0.50	0.0117	0.117	59	99	0.987

**Table 6 molecules-24-00724-t006:** Composition of membrane casting mixture.

Membrane Code	Total Composition (wt %)						Evaporation Time (min)
	Polymers		Additives			Solvent	
	PVDF	PMMA	PEG-200	PVP-K17	TiO2	TEP	
**M0**	6	6	0	0	0	88	2.5
**M1**	6	6	25	5	0	58	0
**M2**	6	6	25	5	0	58	2.5
**M3**	6	6	25	5	0	58	5
**M4**	5	5	25	5	0	60	2.5
**M5**	7	7	25	5	0	56	2.5
**M6**	6	6	25	5	0.12	57.88	2.5
**M7**	6	6	25	5	0.25	57.75	2.5
**M8**	6	6	25	5	0.50	57.50	2.5

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
