# Peer review of "Preparation and Characterization of TiO2-PVDF/PMMA Blend Membranes Using an Alternative Non-Toxic Solvent for UF/MF and Photocatalytic Application"

_molecules, 2019, doi:10.3390/molecules24040724_

Reviewer 1 Report

The authors describe the preparation of mixed matrix membranes using a blend of two polymers (PMMA and PVDF) and TiO2 particles.

The plan of the article has been done well. The authors carry out a systematic study on factors affecting the porosity, hydrophilicity and mechanical stability of the prepared membranes. Moreover they use the photocatalytic propety of the TiO2 nanoparticles to reduce the fouling and polarization concentration.

The main problem with the article is that the authors do not clearly state what they would like to achieve by blending PVDF with PMMA. In the introduction they state that additives such as PVP, PEG and PMMA increase the hydrophilicity of the membranes (page 3, line 108-109). PVP and PEG are hydrophilic but PMMA is hydrophobic! So their statement is incorrect. This point need to be corrected and authors need to explain why they chose to add PMMA to their formulation.

They also need to explain better why and how TEP is a better solvent. Dissolution of PVDF and PMMA in TEP should be studied separately and the results should be compared with the common solvents (such as NMP) for PVDF and PMMA. The solutions of each polymer and their blends should be analyzed by DLS to ensure their solubility. 

Page 9, line 296 the authors state that PMMA is hydrophobic (which is correct) but it contradicts their statement in the introduction where they claim that PMMA is added to the dope solutions to make the PVDF membranes hydrophilic.

Page 10, Line 313 to 317; Authors should explain why and how the membranes are different. Just stating that they are different is not very constructive!

Page 12, line 367-368 please correct the text. This sentence doesn’t make any sense.!

"PMMA chains entangle with PVDF ones and the repulsive interaction between hydrophilic TiO2 and hydrophobic PVDF is weakened"

Page 12, Line 380-382; How does TiO2 act as a crosslinker? Please include an explanation. 

Authors should also include the full characterization of the TiO2 nanoparticles

Author Response

Reviewer 1

The authors describe the preparation of mixed matrix membranes using a blend of two polymers (PMMA and PVDF) and TiO2 particles.

The plan of the article has been done well. The authors carry out a systematic study on factors affecting the porosity, hydrophilicity and mechanical stability of the prepared membranes. Moreover they use the photocatalytic propety of the TiO2 nanoparticles to reduce the fouling and polarization concentration.

Reviewer:

1. The main problem with the article is that the authors do not clearly state what they would like to achieve by blending PVDF with PMMA. In the introduction they state that additives such as PVP, PEG and PMMA increase the hydrophilicity of the membranes (page 3, line 108-109). PVP and PEG are hydrophilic but PMMA is hydrophobic! So their statement is incorrect. This point need to be corrected and authors need to explain why they chose to add PMMA to their formulation.

Authors:

We thank the reviewer for the comment.

PVDF/PMMA blend has been widely studied by several authors. PVDF/PMMA is, in fact, a very well known blend used for improving the performance of polymeric materials (such as processability and conductivity of PVDF). Our aim, in this paper, was to systematically study, for the first time, the effect of some operating conditions (such as exposure time to humidity, presence of pore former additives, polymer concentration) on membrane properties and performance. Moreover, for the first time, we also evaluated the possibility of incorporating TiO2 nanoparticles into PVDF/PMMA membranes and to investigate their catalytic performance. 

In order to better explain the motivation and the importance of our work we, therefore, added the following sentence in the introduction (page 3 line 108):

The blend of PVDF with PMMA was investigated by several authors due to the possibility to combine the benefits of both polymers (improving mechanical resistance, processability and ionic conductivity) [45–48]. However, a systematic study aiming to evaluate the effect of preparation parameters (such as evaporation time, polymer concentration and presence of pore formers) is still missing. Moreover, the potential photocatalytic activity of PVDF/PMMA membranes loaded with TiO2 nanoparticles is an unexplored area.

Considering that photocatalytic composite membranes using alternative solvents have not been thoroughly investigated, the preparation of nano-composite blend PVDF/PMMA membranes containing TiO2 using a more compatible solvent is reported and discussed in this work for the first time. “

Regarding the blend with PMMA it is true that it is a hydrophobic polymer. We never claimed PMMA as a hydrophilic polymer or that PMMA improved the hydrophilicity of PVDF (we just said the opposite at page 3 line 108 where we referred to an article where the blend of PMMA with PVDF improved the hydrophilicity of pure PMMA membranes). In a previous paragraph (page 3 line 95) we stated that several techniques were used for improving PVDF properties such as the blending (without referring to any particular polymer or additive). However, in order to clarify the meaning we modified the paragraph as follows (page 3 line 94):

Many studies have been devoted in improving or varying PVDF membranes properties adopting several techniques such as physical blending, chemical grafting and surface modifications [33–35].

Among these various methods, the blending with various polymers has the advantage of the easy preparation by phase inversion technique.”

Following, we claimed that PVDF can be blended with different polymers and this is particular effective for oxygen containing polymers like PMMA (without referring to them as hydrophilic agents). In particular (page 3 line 101):

“PVDF is highly miscible with oxygen containing polymers, thanks to the interaction between the fluorine atoms and carbonyl groups of the partner polymer [36,37]. Several pairs of blends have been investigated, such as PVDF/poly(vinylpyrrolidone) (PVP), PVDF/ poly (ethylene glycol) (PEG),  PVDF/ poly(vinylacetate) and PVDF/poly(methylmethacrylate) (PMMA) [38]. Among these polymers, PMMA is one of the most  interesting due to its good compatibility with PVDF by simultaneous precipitation in a water bath all over the concentration range in blend polymer [37,39–43].”

Reviewer:

2. They also need to explain better why and how TEP is a better solvent. Dissolution of PVDF and PMMA in TEP should be studied separately and the results should be compared with the common solvents (such as NMP) for PVDF and PMMA. The solutions of each polymer and their blends should be analyzed by DLS to ensure their solubility. 

Authors:

TEP is considered as a greener alternative to the traditional solvents normally used for membrane preparation (like NMP). This is due to its ecotoxicological profile. In the optic of finding approaches for the preparation of membranes which can be considered more environmentally friendly, this solvent can give a great contribute (opposite than NMP). Therefore, as suggested from the referee, we added the following sentence in the introduction (page 3 line 118):

In order to preserve the environment from the risks posed by conventional toxic solvents, triethyl phosphate (TEP) has been, therefore, used as much less harmful alternative solvent, in accordance to the concept of sustainable development and green chemistry principles [45]. TEP, in fact, is not teratogenic, carcinogenic and mutagenic and as stated in its Safety Datasheet “contains no components considered to be either persistent, bioaccumulative and toxic, or very persistent and very bioaccumulative at levels of 0.1% or higher [46]”.

The dissolution of just PVDF in TEP solvent has been already widely investigated by the same authors and also by other colleagues  in other previous publications:

1.      “T. Marino, F. Russo, A. Figoli, The Formation of Polyvinylidene Fluoride Membranes with Tailored Properties via Vapour/Non-Solvent Induced Phase Separation, Membranes 2018, 8, 71”

2.         “S. Fadhil, T. Marino, H.F. Makki, Q.F. Alsalhy, S. Blefari, F. Macedonio, E. Di Nicolò, L. Giorno, E. Drioli, A. Figoli, Novel PVDF-HFP flat sheet membranes prepared by triethyl phosphate (TEP) solvent for direct contact membrane distillation, Chem. Eng. Process. Process Intensif. 102 (2016) 16–26”.

3.        J. Chang, J. Zuo, L. Zhang, G.S. O’Brien, T.S. Chung, Using green solvent, triethyl phosphate (TEP), to fabricate highly porous PVDF hollow fiber membranes for membrane distillation, J. Memb. Sci. (2017).

The dissolution of pure PMMA in TEP has been evaluated but since pure PMMA membranes are fragile with very low mechanical properties, they are not suitable for our application. This makes very difficult the characterization. We do not think it is necessary to perform DLS since visually since the solution of each membrane investigated is homogeneous with not visible particles unsolved.

Reviewer:

3. Page 9, line 296 the authors state that PMMA is hydrophobic (which is correct) but it contradicts their statement in the introduction where they claim that PMMA is added to the dope solutions to make the PVDF membranes hydrophilic.

Authors:

We thank the reviewer for the comment. Indeed, it is true: PVDF and PMMA are both hydrophobic polymers and the contact angle results that we got are in line with the expected results (contact angle higher than 90°).

In the introduction, we stated that the blend PVDF/PMMA produces membranes which are less hydrophobic than the membranes prepared with pure PMMA. It was, therefore, not mentioned that the addition of PMMA makes PVDF more hydrophilic since we just claimed the opposite (citing literature data). In particular (Page 3 line 104):

“Among these polymers, PMMA is one of the most  interesting due to its good compatibility with PVDF by simultaneous precipitation in a water bath all over the concentration range in blend polymer [37,39–43]. The produced blend membranes exhibited improved hydrophilicity and water permeation fluxes with respect to pure PMMA membranes [44].”

Reviewer:

4. Page 10, Line 313 to 317; Authors should explain why and how the membranes are different. Just stating that they are different is not very constructive!

Authors:

We thank the reviewer for the comment. We agree with the reviewer and we added in the manuscript the motivation for the difference in membrane structure citing the appropriate references (page 10 line 327):

The reduction of cavities as a consequence of polymer concentration increase is a very well documented phenomenon [57,64]. High polymer concentrations, in fact, slow down the precipitation process of the membrane leading to the formation of a membrane with a denser top layer and lower pore size [65].”

Reviewer:

5. Page 12, line 367-368 please correct the text. This sentence doesn’t make any sense.!

"PMMA chains entangle with PVDF ones and the repulsive interaction between hydrophilic TiO2 and hydrophobic PVDF is weakened"

Authors:

We thank the reviewer for the comment. The sentence that was not clear was removed and completely reformulated (pag. 12 line 382):

“…No clear particle aggregation was observed in the cross sections of the composites, indicating that the TiO2 nanoparticles were well dispersed in the polymer matrix. These results are similar to the ones reported by Zho et al. [77] for PVDF/PMMA/TiO2…….”

Reviewer:

6. Page 12, Line 380-382; How does TiO2 act as a crosslinker? Please include an explanation. 

Authors: We thank the reviewer for the comment. We added the explanation  why TiO2 acts as a cross-linking agent towards many polymers. TiO2, in fact, is able to form hydrogen bonds with polymer chains increasing their rigidity. We added the explanation in the manuscript (page 12 line 391):

Moreover, the addition of TiO2 led to a general improvement in membrane Young’s modulus value in comparison to the unloaded membrane M2. As reported in several works [78,80,81], in fact, the introduction of TiO2 nanoparticles enhances the mechanical resistance of the membranes by acting as cross-linkers between the polymer chains increasing their rigidity and, as a consequence, their mechanical resistance. The TiO2 nanoparticles can, in fact, interact via hydrogen bond between the hydroxyl group present on the surface of TiO2 and oxygen groups of PMMA [82,83].”

Reviewer:

7. Authors should also include the full characterization of the TiO2 nanoparticles

Authors:

The TiO2 nanoparticles that we used are commercial as reported in materials and methods sections. The full characterization of TiO2 nanoparticles is, therefore, reported in their datasheet (and on the web) and this is the reason why we decided to not add this info in the manuscript.

In particular, here we report their characterization info (which we do not think it is needed to be added in the manuscript):

AEROXIDE® TiO2 P25 is a fine white powder with hydrophilic character caused by hydroxyl groups on the surface. It consists of aggregated primary particles. The aggregates are several hundred nm in size and the primary particles have a mean diameter of approx. 21 nm. Particle size and density of ca. 4 g / cm³ lead to a specifi c surface of approx. 50 m²/g. Due to the formation of aggregates and agglomerates, the tamped density of AEROXIDE® TiO2 P25 is only about 130 g / l (determined acc. to DIN ISO 787 / XI). The weight ratio of anatase and rutile is approximately 80 / 20. Both crystal forms are tetragonal but with different dimensions of the elementary cell. At 300 °C, a slow conversion of anatase to the more stable rutile structure begins. At temperatures higher than 600 °C, the conversion runs faster combined with a reduction of the specific surface. AEROXIDE® TiO2 P25 is suitable for many applications that require a high photoactivity.”

Reviewer 2 Report

This manuscript describes the photocatalytic porous membrane made of PVDF/PMMA blended with TiO2 nanoparticles by the phase inversion method using triethyl phosphate (TEF), which is an alternative solvent. Authors carefully study the membrane properties including porous structure, mechanical strength, permeability and so on as a function of blending ratio of each component, evaporation time and another important variant. Therefore, I recommend this work publish in the Molecules, but only after the authors have successfully addressed a few key aspects of the manuscript detailed below, in particular, 1 and 2. 1. Authors measured the pure water permeability (PWP) and checked the photocatalytic performance by simply immersing the membrane in the MB solution without permeation. However, since the photocatalytic performance was not measured during membrane permeation, I think that the performance as a photocatalytic membrane was not measured. Therefore, I think the author should confirm the photocatalytic performance during permeation the MB solution through the membrane in order to call it the photocatalytic membrane. (c.f. Sci. Rep. 7, 3128, 2017; Ind. Eng. Chem. Res., 52 (39), 13938, 2013). 2. I’m concerning about the stability of the membranes, which is consist of PVDF/PMMA blend. In particular, PMMA is a well known photo-decomposable polymer (e.g. J. Phys. Chem. 65(6), 967, 1961; J. Polym. Sci. A 4(5), 1209, 1966), and thus I think there will be severe photodecomposition during the UV irradiation for the photocatalytic process. Therefore, I think authors should mention this issue and show the stability of the polymer membrane during the photocatalytic process. 3. I enjoyed to read the introduction and I feel like authors successfully introduce the history of their research field (e.g. photocatalytic membrane). However, your introduction part could be improved by proper citation of previous works. I saw many descriptions about previous works without proper citation in your introduction part. For instance, in page 3, line 118, authors are mentioning the 4 previous works, which have used TEP for the production of PVDF membranes, but there is no citation for the previous works. If you mentioned important previous works directly related with your paper, you should cite the previous work in your introduction. Therefore, I recommend you to check your introduction part and add proper references. 4. The experimental section should be written in more detail. In particular, I think the UV irradiation condition is quite important information during the photocatalytic experiment, however, there is no wavelength of UV, power and so on. Therefore, I recommend you to describe more detail in the experiment section. 5. When you want to use abbreviations, the full name should be mentioned first. However, for "MF" the first mention is on page 7, but the full name is mentioned first on page 9. 6. I think the cross-sectional SEM images are really important to check the porous structure and texture of the overall membrane structure. But I feel like the cross-sectional images in Figure 4, 6, and 7 are too small and low magnified to check the detailed porous structures and textures. I suggest you add higher magnified cross-sectional images.

Author Response

Reviewer 2

This manuscript describes the photocatalytic porous membrane made of PVDF/PMMA blended with TiO2 nanoparticles by the phase inversion method using triethyl phosphate (TEF), which is an alternative solvent. Authors carefully study the membrane properties including porous structure, mechanical strength, permeability and so on as a function of blending ratio of each component, evaporation time and another important variant. Therefore, I recommend this work publish in the Molecules, but only after the authors have successfully addressed a few key aspects of the manuscript detailed below, in particular, 1 and 2.

Reviewer:
1. Authors measured the pure water permeability (PWP) and checked the photocatalytic performance by simply immersing the membrane in the MB solution without permeation. However, since the photocatalytic performance was not measured during membrane permeation, I think that the performance as a photocatalytic membrane was not measured. Therefore, I think the author should confirm the photocatalytic performance during permeation the MB solution through the membrane in order to call it the photocatalytic membrane. (c.f. Sci. Rep. 7, 3128, 2017; Ind. Eng. Chem. Res., 52 (39), 13938, 2013).

Authors:

We thank the reviewer for the comment. Indeed, we did not measure the photocatalytic activity of the membrane during the filtration process. However, the primary aim of this paper was to systematically study, for the first time, the effect of some operating conditions (such as exposure time to humidity, presence of pore former additives, polymer concentration) on the properties and performance of PVDF/PMMA membranes. Moreover, for the first time, we also evaluated the possibility of adding TiO2 nanoparticles in PVDF/PMMA membranes and to investigate their influence on membrane properties and performance. The photocatalytic tests performed can be considered just as a preliminary investigation in order to prove:

-          The resistance of the blend PVDF/PMMA membrane to UV light irradiation

-          The activity of the catalysts embedded in the PVDF/PMMA in the degradation of organic pollutants (like methylene blue)

We are aware that for a full investigation of membrane catalytic performance the photocatalytic tests have to be carried out during the operation time of the membrane. This topic will be a key objective of a future work. To better clarify this point, we added the following sentence in the manuscript (page 15  line 458):

“These preliminary results can be considered as a starting point for a future far-reaching investigation of membranes photocatalytic activity directly measured during the operation time of the membrane (during filtration) considering, also, different organic pollutants.”

Reviewer:
2. I’m concerning about the stability of the membranes, which is consist of PVDF/PMMA blend. In particular, PMMA is a well known photo-decomposable polymer (e.g. J. Phys. Chem. 65(6), 967, 1961; J. Polym. Sci. A 4(5), 1209, 1966), and thus I think there will be severe photodecomposition during the UV irradiation for the photocatalytic process. Therefore, I think authors should mention this issue and show the stability of the polymer membrane during the photocatalytic process.

Authors:

We thank the reviewer for the precious comment. During the exposure of PVDF/PMMA membranes to UV light irradiation we did not observe any degradation or alteration of the membrane.

However, in order to fully exclude the possibility of a photodegradation of the blend membranes we carried out a further experiment. We measured the water permeability of the PVDF/PMMA blend membrane containing the highest concentration of TiO2 (0.5 wt%). Once recorded, we exposed the same membrane sample to UV irradiation (the same used for photocatalytic tests) for 3h. After this period, the water permeability was tested again. We did not observe any variation in water permeability before and after the UV treatment. As a consequence of a possible photodegradation of the PVDF/PMMA membrane, in fact, an increase in water permeability should have been observed.

We added, therefore, to the manuscript the following sentence (page 13  line  408)

Before carrying out the photocatalytic tests with MB, the resistance of the blend membranes to UV irradiation was evaluated in order to exclude any possible photodegradation of the polymer matrix. The tests were carried out with the M8 membrane, containing the highest concentration of TiO2, by measuring its starting water permeability. Then, the membrane sample was exposed for 3h to UV light irradiation. After this period, the water permeability was measured again. No changes in the recorded values of water permeability, before and after the exposure to UV light irradiation, were observed. From the constancy of membrane performance, the resistance of the membrane to UV light was, therefore, assessed. In case of photodegradation of the polymer, in fact, an increase in water permeability should have been observed.”

Moreover, we found an article where PMMA electrospun membranes containing TiO2 were also successfully used for the photodegradation of methylene blue without any problem related to membrane degradation [1].

Ref.

[1]   Vild, A., Teixeira, S., Kuhn, K., Cuniberti, G. & Sencadas, V. (2016). Orthogonal experimental design of titanium dioxide - Poly(methyl methacrylate) electrospun nanocomposite membranes for photocatalytic applications. Journal of Environmental Chemical Engineering, 4 (3), 3151-3158.

Reviewer:
3. I enjoyed to read the introduction and I feel like authors successfully introduce the history of their research field (e.g. photocatalytic membrane). However, your introduction part could be improved by proper citation of previous works. I saw many descriptions about previous works without proper citation in your introduction part. For instance, in page 3, line 118, authors are mentioning the 4 previous works, which have used TEP for the production of PVDF membranes, but there is no citation for the previous works. If you mentioned important previous works directly related with your paper, you should cite the previous work in your introduction. Therefore, I recommend you to check your introduction part and add proper references.

Authors: We thank the reviewer for the comment. We agree with the reviewer and we added in the manuscript the missing references.

In particular, page 3 line 124:

So far, there are only five articles concerning the use of TEP, as only solvent, for the production of PVDF membranes  applied in membrane distillation but none of them dealing with water purification [47–51].”

Reviewer:
4. The experimental section should be written in more detail. In particular, I think the UV irradiation condition is quite important information during the photocatalytic experiment, however, there is no wavelength of UV, power and so on. Therefore, I recommend you to describe more detail in the experiment section.

Authors: We thank the reviewer for the comment. The details of the lamp including emission and power have been added (pag. 6 line 208):

A batch photocatalytic reactor consisted of a pyrex glass tank with a piece of membrane was placed inside a shielded chamber with UV lamp mounted on the top (lamp emission from 180 nm to visible light- 500W– Purchased from Helios Italquarz s.r.l.).”

Reviewer:
5. When you want to use abbreviations, the full name should be mentioned first. However, for "MF" the first mention is on page 7, but the full name is mentioned first on page 9.

Authors: We thank the reviewer for the comment. We revised the abbreviations throughout the whole manuscript.

Reviewer:
6. I think the cross-sectional SEM images are really important to check the porous structure and texture of the overall membrane structure. But I feel like the cross-sectional images in Figure 4, 6, and 7 are too small and low magnified to check the detailed porous structures and textures. I suggest you add higher magnified cross-sectional images.

Authors: We thank the reviewer for the comment. As requested by the referee, we performed new SEM images at higher magnification. We created a supporting material file where we added, for all the investigated membranes, the SEM pictures with magnification of cross-sections.

Round  2

Reviewer 1 Report

The authors have answered majority of my previous questions. The paper can be accepted after addressing the following points;

Regarding question 2 the authors need to add the information on the dissolution of PVDF in TEP to the manuscript and add the relevant references.

On the use of DLS, I completely disagree with the authors. Visual observation of dissolution is no proof of molecular dissolution especially when macromolecules are concerned. DLS data should be included.

The information regarding the TiO2 particles provided by the manufacturer should be added to the materials and methods section. 

Author Response

Regarding question 2 the authors need to add the information on the dissolution of PVDF in TEP to the manuscript and add the relevant references. On the use of DLS, I completely disagree with the authors. Visual observation of dissolution is no proof of molecular dissolution especially when macromolecules are concerned. DLS data should be included.

Authors:

We thank the editor for the comment. We included the previous works where the solubility of PVDF in TEP was studied in terms of Hansen solubility parameters (Page 6 line 225):

“This work describes, for the first time, the preparation of PVDF/PMMA membranes using TEP as a solvent. However, the ability of TEP to dissolve PVDF has been already studied in previous works through the comparison of their Hansen solubility parameters [52,56]. The solubility parameter of TEP (22.2 MPa1/2), in fact, is very close to the one of PVDF (23.2 MPa1/2) confirming that a PVDF-TEP solution is thermodynamically favored.”

 On the use of DLS, unfortunately we do not have the equipment in our institute and we are not able to perform the requested measurements.

The information regarding the TiO2 particles provided by the manufacturer should be added to the materials and methods section. 

Authors:

We thank the editor for the comment. We added the following summarized information in Materials and Methods Section (page 3 line 139):

“As reported by the supplier, the TiO2 nanoparticles have a primary mean diameter of about 21 nm, with a density of about 4 g/cm3 with a predominance of the anatase form”.

Reviewer 2 Report

This manuscript describes the photocatalytic porous membrane made of PVDF/PMMA blended with TiO2 nanoparticles by the phase inversion method using triethyl phosphate (TEF), which is alternative solvent. As I suggested, authors have carefully revised their manuscript, except for a few major parts. I think this manuscript will be able to be accepted after some minor modification.

1. From my concerns about the stability of the membrane during the UV irradiation, the authors added the paragraph (Page 13, line 412) about the resistance of the membranes to UV irradiation. It seems reasonable, however, there is the only explanation, and no data to support their argument. I think the author should add the stability test data (The permeability test before/after UV irradiation) to the supporting materials to support their argument.

Author Response

We thank the editor for the comment. We added a further graph in Supporting materials showing the water permeability before and after UV irradiation. In addition, in the manuscript we modified the text accordingly (page 14 line 415):

Before carrying out the photocatalytic tests with MB, the resistance of the blend membranes to UV irradiation was evaluated in order to exclude any possible photodegradation of the polymer matrix. The tests were carried out with the M8 membrane, containing the highest concentration of TiO2, by measuring its starting PWP. Then, the membrane sample was exposed for 3h to UV light irradiation. After this period, the PWP was measured again. No changes in the recorded values of PWP (about 780 L/m2 h bar), before and after the exposure to UV light irradiation, were observed (see Fig. S8 of the Supporting Material). From the constancy of membrane performance, the resistance of the membrane to UV light was, therefore, assessed. In case of photodegradation of the polymer, in fact, an increase in water permeability should have been observed. The higher PWP of M8 membrane in comparison to M1, M2 and M3 can be attributed to its higher pore size and lower contact angle as a consequence of TiO2 addition.”